# The Role of Wild Boars in the Circulation of Tick-Borne Pathogens: The First Evidence of *Rickettsia monacensis* Presence

**DOI:** 10.3390/ani13111743

**Published:** 2023-05-24

**Authors:** Ioana Adriana Matei, Zsuzsa Kalmár, Anamaria Balea, Marian Mihaiu, Attila D. Sándor, Adrian Cocian, Smaranda Crăciun, Cosmina Bouari, Violeta Tincuța Briciu, Nicodim Fiț

**Affiliations:** 1Department of Microbiology, Immunology and Epidemiology, Faculty of Veterinary Medicine, University of Agricultural Sciences and Veterinary Medicine, 400372 Cluj-Napoca, Romania; 2Department of Infectious Diseases, “Iuliu Hațieganu” University of Medicine and Pharmacy, 400012 Cluj-Napoca, Romania; 3ELKH-ÁTE Climate Change, New Blood-Sucking Parasites and Vector-Borne Pathogens Research Group, 1078 Budapest, Hungary; 4Sanitary Veterinary and Food Safety Directorate Cluj, 400609 Cluj-Napoca, Romania; 5Department of Animal Breeding and Food Science, Faculty of Veterinary Medicine, University of Agricultural Sciences and Veterinary Medicine, 400372 Cluj-Napoca, Romania; 6Department of Parasitology and Parasitic Diseases, Faculty of Veterinary Medicine, University of Agricultural Sciences and Veterinary Medicine, 400372 Cluj-Napoca, Romania; 7Department of Parasitology and Zoology, University of Veterinary Medicine, 1078 Budapest, Hungary

**Keywords:** *Anaplasma* spp., *Rickettsia* spp., *Sus scrofa*, sentinel species, spotted fever, Romania

## Abstract

**Simple Summary:**

Wildlife has a major role in the transmission and maintenance of zoonotic agents, as most emerging infectious diseases are of wildlife origin. Among the great variety of wildlife species, wild boars (*Sus scrofa*) are species with continuously increasing abundance and geographical distribution. Wild boars were suggested as appropriate sentinel species for the distribution and abundance of certain tick species in America and Europe. Therefore, these species may also be suitable sentinel species for screening for tick-borne pathogens. The aim of this study was to evaluate the presence of tick-borne pathogens in wild boars in Romania. A total of 203 blood samples were collected from wild boars (*Sus scrofa*) from two Transylvanian counties (Cluj, Sălaj) in Romania. Three emerging zoonotic pathogens were identified, *Anaplasma phagocytophilum* and two *Rickettsia* species (*R. helvetica*, *R. monacensis*) DNA, in the blood of tested animals. These results suggest the involvement of wild boars in the epidemiology of *Rickettsia* species and *A. phagocytophilum*. Hence, this species may be used as a sentinel for the general surveillance of these zoonotic pathogens.

**Abstract:**

Most wild mammals can serve as hosts both for tick-borne pathogens (TBPs) and for the ticks themselves. Among these, wild boars, due to their large body size, habitat and life span, show high exposure to ticks and TBPs. These species are now one of the widest-ranging mammals in the world, as well as the most widespread suid. Despite the fact that certain local populations have been decimated by African swine fever (ASF), wild boars are still considered overabundant in most parts of the world, including Europe. Altogether, their long-life expectancy, large home ranges including migration, feeding and social behaviors, wide distribution, overabundance and increased chances of interactions with livestock or humans make them suitable sentinel species for general health threats, such as antimicrobial-resistant microorganisms, pollution and ASF geographical distribution, as well as for the distribution and abundance of hard ticks and also for certain TBPs, such as *Anaplasma phagocytophilum*. The aim of this study was to evaluate the presence of rickettsial agents in wild boars from two counties in Romania. Among 203 blood samples of wild boars (*Sus scrofa* ssp. *attila*) collected during 3 (2019–2022) hunting seasons (September–February), 15 were found positive for tick-borne pathogen DNA. Six wild boars were positive for *A. phagocytophilum* DNA presence and nine for *Rickettsia* spp. The identified rickettsial species were *R. monacensis* (six) and *R. helvetica* (three). No animal was positive either for *Borrelia* spp., *Ehrlichia* spp. or *Babesia* spp. To the best of our knowledge, this is the first report of *R. monacensis* in European wild boars, thus adding the third species from the SFG *Rickettsia,* in the epidemiology of which this wild species may have a role as a reservoir host.

## 1. Introduction

Wildlife has a major role in the transmission and maintenance of zoonotic agents, as most emerging infectious diseases are of wildlife origin [1]. Among the great variety of wildlife species, ungulates, including wild boars (*Sus scrofa*), are species with continuously increasing abundance and geographical distribution [2]. The increase in the range and population of wild boars and the causes leading to this have been intensely debated among different scientific fields. However, the most accepted causes include climate change, together with the low mortality of piglets during the winter and the decreasing populations of natural predators [2]. Wild boars are an endemic species in all of Europe and Asia and an invasive species in Northern Africa; North, Central and South America; Australia; and New Zealand [3]. Concerning their abundance, they are considered overabundant in most agricultural and rural areas in Europe, as well as in many peri-urban areas [2]. Wild boars are versatile species living in a large variety of habitats, the only limitation being the presence of water. They are omnivore animals and their average life span is eight to ten years [4]. The home range is highly variable depending on multiple factors, including group structure, resources and accessibility. In the absence of the dominating sow (old matriarch), groups can travel long distances, covering up to 1000 ha in a day, sometimes reaching anthropic zones, including peri-urban areas, in the search for food [4]. Among the multiple effects of their overabundance, together with the abovementioned ecological and biological characteristics, it is included the increasing interaction of wild boars with domestic animals and even with humans, with possible negative impacts on animal and human health. Wild boars are important reservoirs and disseminators of infectious agents, such as African swine fever virus, classical swine fever, and Aujeszky’s disease virus, affecting livestock, and also for zoonotic pathogens, such as hepatitis E virus, *Brucella suis*, and *Mycobacterium tuberculosis* [5]. In addition to these, many other zoonotic infectious agents have been detected in wild boars, without knowing well the ecological role and importance of these animals in their epidemiology [5]. Due to their ecological and biological characteristics, wild boars have been suggested as appropriate sentinel species for hard tick distribution and abundance in America [6] and Europe [7]. Although not fully evaluated, they may also be suitable sentinel species for the screening of tick-borne pathogens (TBP), in addition to already suggested groups, such as red foxes or dogs [8]. 

Tick-borne diseases (TBDs) represent around 95% of all vectorial disease cases in North America, Europe and Asia [9]. Ticks are hematophagous arthropods with a complex life cycle and variable host specificity depending on the species and developmental stage [9]. In Europe, in general, and in Romania, in particular, the most prevalent genera are *Ixodes*, *Dermacentor*, *Rhipicephalus*, *Haemaphysalis* and *Hyalomma* [10]. Altogether, these ticks may transmit a large variety of TBPs. The commonly reported emergence of TBD is caused by numerous factors, including climate change and the abundance and accessibility of available hosts [11]. The importance of TBD for public and animal health instills major importance for the surveillance of ticks in the continuous effort to control and limit their impact. An important tool in disease surveillance is the monitoring of agents present in the sentinel and reservoir species. Among diverse TBPs, wild boars have been suggested as suitable reservoir species for *Anaplasma phagocytophilum*, with multiple reports of this bacteria in boars in Europe, including Romania [12]. Concerning other TBPs, in addition to serological studies, *Borrelia burgdorferi* s.l. has been molecularly detected in wild boars in Italy and Portugal [13,14]. Recently, *Babesia divergens* was also molecularly detected in this animal species in the Czech Republic [15]. Reports of *Rickettsia* spp. in European wild boars are few, with only two species being recorded so far: *R. slovaca* in Italy and *R. helvetica* in the Netherlands [16,17,18,19]. In addition, *R. slovaca* was also identified in the spleen samples of wild boars in Algeria, North Africa [20]. Nevertheless, there are several European studies reporting a high prevalence of spotted fever group *Rickettsia* (SFG *Rickettsia*) in ticks collected from wild boars, coupled with negative spleen or blood samples in hosts [21,22]. However, the high seroprevalence of *Rickettsia* spp. reported in Spain, Brazil and Japan may suggest their involvement in rickettsiosis epidemiology [23,24,25,26]. Although there are certain differences between wild boar subspecies across the globe (including between European countries), and also among tick populations, it is expected to have similar observations in Romania. 

Considering the potential role of wild boars as hosts and sentinel species for tick-borne diseases, in general, and the previous detection of some TBPs, together with contradictory results of *Rickettsia* spp. detection across the world, the aim of this study was to evaluate the presence of TBPs in wild boars in Romania. 

## 2. Materials and Methods

### 2.1. Sampling

Currently, between 6 and 16 wild boar subspecies are recognized worldwide. In Europe, three subspecies are present, and in Romania, there are two: *Sus scrofa* ssp. *scrofa* and *S. s. attila* [27]. For subspecies identification, the most commonly used measurements are lacrimal bone shape and size, molar size and total body weight [27,28,29]. In Transylvania (study’s location), only Eastern European *S. s. attila* is present. *Sus scrofa attila* is a large-sized subspecies, which can exceed 350 kg weight, with long lacrimal bones and dark hair, usually lighter colored and larger in size than *S. s. scrofa* [27,28,29].

The samples were collected during three hunting seasons during 2019–2022. Wild boars were shot by authorized local hunters during the regular hunting season (October–February) as trophies, for population control or for conflict management purposes. The measurements of total body weight and third molars were recorded. The external examination of wild boar carcasses was performed in order to collect the ticks (if present), and a blood sample was collected also. A total of 54 blood samples of boars (*S. s. attila*) were collected during 2 hunting seasons (2019–2020 and 2020–2021) in Sălaj county in one particular hunting area, representing all the animals hunted in the chosen hunting seasons. Unfortunately, because of African swine fever (ASF) in the first sampling area (high reported mortality and, consequently, an important decrease in the population), in the third hunting season (2021–2022), no blood samples were available. Because of this, 149 additional blood samples collected by local hunters in the mentioned time period and sent to the local veterinary authority from Cluj County (a neighboring county) for ASF surveillance were included in the study. 

### 2.2. Molecular Analysis

Genomic DNA was extracted individually from each sample using ISOLATE II Genomic DNA Kit (Meridian Bioscience, Newtown, OH, USA) following the manufacturer’s instructions. 

The presence of TBP’s DNA was evaluated by conventional and nested PCR amplifying fragments of the 16S rRNA genes of *A. phagocytophilum* and *Ehrlichia* spp., 18S rRNA gene of *Babesia*/*Theileria* spp., intergenic spacer (IGS) of *Borrelia* spp. and 17 kDa outer membrane antigen gene of *Rickettsia* spp. *Anaplasma*-positive samples were further evaluated by heminested PCR amplifying fragments of the *groEL* gene of *A. phagocytophilum* (Table 1). 

In all cases, the amplification was performed in 25 µL of reaction mixture containing 12.5 µL of 2 × MyTaqTM Red Mix (Bioline, London, UK), 6.5 µL of PCR water, 1 µL of each primer (0.01 mM) and 4 µL aliquot of isolated DNA, and 1 µL of the primary PCR products instead of DNA template for the nPCR.

The amplification profile for *Anaplasma* spp./*A. phagocytophilum* was the following: the first PCR consisted of initial denaturation at 95 °C for 5 min, followed by 40 cycles of denaturation at 94 °C for 30 s, annealing at 55 °C for 30 s and extension at 72 °C for 60 s, ending with a final extension at 72 °C for 5 min. For the second PCR, the amplification profile consisted of initial denaturation at 95 °C for 5 min, followed by 30 cycles of denaturation at 94 °C for 30 s, annealing at 55 °C for 30 s and extension at 72 °C for 60 s, ending with a final extension at 72 °C for 5 min. For *groEL*, the first PCR consisted of initial denaturation at 95 °C for 5 min, followed by 35 cycles of denaturation at 95 °C for 30 s, annealing at 51 °C for 45 s and extension at 72 °C for 90 s, ending with a final extension at 72 °C for 5 min. The nested step followed the same profile with an annealing temperature of 55 °C.

The amplification profile for *Babesia*/*Theileria* spp. was the following: the first PCR consisted of initial denaturation at 95 °C for 10 min, followed by 40 cycles of denaturation at 95 °C for 30 s, annealing at 54 °C for 30 s and extension at 72 °C for 40 s, ending with a final extension at 72 °C for 5 min.

The amplification profile for *Borrelia* spp. was the following: 15 min at 90 °C, followed by cycles of 20 s at 90 °C, 30 s at 70 °C, 30 s at 72 °C, lowering the annealing temperature 1 °C/cycle until reaching 60 °C and, finally, 40 cycles at 60 °C, followed by 7 min at 72 °C.

The amplification profile for *Ehrlichia* spp./*E. canis* was the following: the first PCR consisted of initial denaturation at 95 °C for 5 min, followed by 35 cycles of denaturation at 94 °C for 30 s, annealing at 64 °C for 30 s and extension at 72 °C for 60 s, ending with a final extension at 72 °C for 5 min. For the second PCR, the amplification profile consisted of initial denaturation at 95 °C for 5 min, followed by 35 cycles of denaturation at 94 °C for 30 s, annealing at 59 °C for 30 s and extension at 72 °C for 60 s, ending with a final extension at 72 °C for 5 min.

For the amplification profile for SFG *Rickettsia* spp. 17 kDa outer membrane gene amplification, the profile consisted of an initial denaturation at 95 °C for 3 min, followed by 35 cycles of denaturation at 95 °C for 30 s, annealing at 61 °C for 30 s, elongation at 72 °C for 45 s, followed by a final elongation at 72 °C for 5 min. The nested step followed the same profile with an annealing temperature of 54 °C.

Positive and negative controls were included in each PCR set to assess the possible cross-contamination and the specificity of the reactions. PCR was carried out using T100^TM^ Thermal Cycler (Bio-Rad, Hercules, CA, USA). Amplicons were visualized by using electrophoresis in 1.5% agarose gel stained with RedSafe^®^ Nucelic Acid Staining Solution (iNtRON Biotechnology, JungAng Induspia, Seongnam-Si, Republic of Korea). 

PCR-positive samples were purified using FavorPrep GEL/PCR Extraction or Purification DNA fragment Kit (Favorgen, Ping Tung, Taiwan) and were further sequenced (Macrogen Europe, Amsterdam, The Netherlands). DNA sequences were analyzed and compared with those available in GenBank^TM^ by using Basic Local Alignments Tool (BLAST) analysis.

### 2.3. Phylogenetical Analysis

The phylogenetic analysis was performed by MEGA11 software for 17 kDa surface antigen coding gene of *Rickettsia* spp. and *groEL* gene of *A. phagocytophilum* using reference sequences of strains for which sequence data were available in GenBank. The phylogenetic trees were constructed using maximum likelihood method with Tamura–Nei model and a Gamma distribution of 500 bootstrap replicates. The accession numbers, country and host of isolates from each species are displayed in the phylogenetic trees.

### 2.4. Statistical Analysis

Statistical analysis was performed using Epi Info^TM^ 7.2 software (CDC, Atlanta, GA, USA). The infection prevalence and its 95% confidence intervals were assessed using the chi-square independence test.

## 3. Results

The overall infection prevalence of TBP in the collected blood samples from wild boars was 7.4% (15/203; 95% CI: 4.2–11.9). None of the blood samples were positive for *Borrelia* spp., *Ehrlichia* spp. or *Babesia*/*Theileria* spp. 

The 16S rRNA sequence of *A. phagocytophilum* DNA was identified in six (203; 3.0%, 95% CI: 1.1–6.3) blood samples. The infection prevalence in Cluj County was 3.4% (5/149; 95% CI: 1.1–7.7), and all of the positive samples were detected in only one of the seasons (2021–2022). While in Sălaj County, the infection prevalence was 1.9% (1/54; 95% CI: 0.1–9.9), and only one positive sample was collected in the first season (2019–2020).

*Anaplasma-phagocytophilum*-positive samples for 16S rRNA were further amplified targeting the *groEL* gene of *A. phagocytophilum*. The sequence analysis revealed 100% similarity among DNA sequences from the present study and the same percentage with *A. phagocytophilum* isolated from *S. scrofa* blood from the Czech Republic (MT498612). Additionally, high similarity (99.78–100%) with various other strains from GenBank isolated from engorged *I. ricinus* in different European countries (MN093180, the Netherlands; AY281823, Germany; OP265401, Czech Republic; MF372791, Hungary; HQ629908, Russia) and dog blood (KT970680, Italy) was shown. According to the nucleotide BLAST analysis, the sequences belong to the ecotype I described by Jahfari et al. 2014 [36]. This ecotype consists of strains with zoonotic potential, with all HGA strains belonging to this ecotype.

A total of 33 available sequences of the *A. phagocytophilum groEL* gene in GenBank (one isolate from each European country/host) was included in the phylogenetic analysis (Figure 1). The ecotype of *A. phagocytophilum* isolates was not included in the phylogenetic tree since this information was not available in GenBank for each sequence. *Anaplasma phagocytophilum* isolated from wild boars in Romania was clustered together with an isolate from red foxes (the Czech Republic; Figure 1).

*Rickettsia* spp. DNA was detected in 4.4% (9/203; 95% CI: 2.1–8.3) of the samples. The infection prevalence in Cluj County was 2.0% (3/149; 95% CI: 0.4–5.8) and 11.1% (6/54; 95% CI: 4.2–22.6) in Sălaj County. Two positive samples (7.7%, 95%CI: 1.0–25.1) were collected in the 2019–2020 season, four were collected (14.28%, 95%CI: 4.0–14.3) in 2020–2021, and three were collected (2.0%, 95% CI: 0.4–5.8) in 2021–2022. We identified two *Rickettsia* species in both counties, *R. monacensis* (*n* = 6) and *R. helvetica* (*n* = 3). 

The sequence analysis using the BLAST of both *Rickettsia* species revealed a high similarity (99.9–100%) among DNA sequences from the present study. *Rickettsia monacensis* showed 99.9–100% similarity to several strains isolated from different hosts in Europe (Lithuania and Italy) identified in ticks that feed on birds (MF491748), sheep (KY319220), roe deer (KY319219) and humans (KY319215). 

Additionally, the sequences of *R. helvetica* recorded in wild boars showed 99.9–100% similarity to several strains isolated from different ticks in Europe (Romania, Lithuania, Italy and Poland): *Ixodes kaiseri* (MK733577), *I. ricinus* from the environment (KT734810) and *I. ricinus* collected from different bird species (MF491752 and MF491751), as well as ungulates (KY346828), but also from mites (MF491777 and MF491768), insects (MF491768) and hosts’ tissues of rodents (MF491760 and MF491758) and insectivore species (MF491754), too. A total of 32 sequences (1 isolate from each European country/host) of *R. helvetica* (*n* = 20), *R. monacensis* (*n* = 8), *R. raoultii* (*n* = 2) and *R. slovaca* (*n* = 2) were included in the phylogenetic tree analysis. *R. monacensis* and *R. helvetica* isolated from wild boars from Romania were clustered together (Figure 2). However, available sequences in GenBank of 17 kDa surface antigen coding gene of different *Rickettsia* species distributed in Europe were scarce or missing for certain *Rickettsia* species (e.g., *R. aeschlimannii*, *R. conorii* and *R. massiliae*).

None of the samples showed mixed infections of different TBP. No ticks were found on the wild boars from Sălaj County, and no information was available regarding tick presence on boars from Cluj County.

## 4. Discussion

Altogether, the DNA of *A. phagocytophilum* and two *Rickettsia (R. helvetica* and *R. monaensis*) species were recorded in wild boar tissues in Romania in the present study. While both *A. phagocitophylum* and *R. helvetica* have already been detected in wild boars [12,13,14,15,16,17], to the best of our knowledge, this is the first report of *R. monacensis* in wild boars worldwide. So far, within SFG rickettsiae, only *R. helvetica* and *R. slovaca* have been detected in wild boar tissue samples in Europe [16,18,19].

*Rickettsia slovaca,* one of the agents of scalp eschar and neck lymphadenopathy after tick bites (SENLAT), was, for the first time, identified in wild boars in Italy in one out of nine tested skin biopsies. In the same study, ticks (*Dermacentor marginatus*) collected from wild pigs showed an SFG rickettsiae prevalence of 33.9%, with 2 species being identified as *R. slovaca* (36 ticks, including 1 collected from wild boar with positive skin biopsy) and *R. raoultii* (2 ticks) [19]. More recently, 11.3% of Italian wild boar tissues (1 liver and 8 ear biopsies out of 80 tissues tested) were found to be positive for SFG rickettsiae, identified as *R. slovaca*. Similar to a previously described study, a high *R. slovaca* prevalence was observed in ticks collected from wild boars, namely, 48% in *D. marginatus* and 33.3% in *D. reticulatus*. In addition, *Candidatus* R. rioja was detected in three *D. marginatus* collected from wild boars, another recently recognized SENLAT agent [18,37]. In another Italian study, 1 out of 93 spleen samples tested positive for *R. slovaca.* Among the ticks collected from these wild boars, 16 (9%) *D. marginatus* were positive for *R. slovaca* and 1 (0.6%) for *R. raoultii*, and a single *I. ricinus* (0.6%) was infected by *R. slovaca* [19]. The detection of *R. slovaca* in Italian wild boars is sustained by the high seropositivity to *R. slovaca* (52.2%) observed in Spanish wild boars [23]. Similar to studies conducted in Italy, *R. slovaca* was also identified in 30.5% of *D. marginatus* ticks removed from wild boars [23]. In addition to the European reports, *R. slovaca* was detected in wild boar spleen samples collected between 2011–2014 in Algeria [20]. 

*Rickettsia helvetica* was detected in two whole blood samples (7%) of boars in the Netherlands [17]. Despite these reports of SFG rickettsiae in wild boar tissue samples, other studies report negative results in tissue samples. For instance, a study in Portugal on 65 animals reported negative results in all blood samples for the following: *Babesia* spp., *B. burgdorferi* (s.l.), *Ehrlichia* spp. and *Rickettsia* spp. [38]. In this study, no data regarding tick presence and TBP in the ticks are provided. In another study conducted in Italy, *D. marginatus* ticks collected from wild boars were positive for *R. slovaca* (24%) and *R. helvetica* (1%)*,* while *I. ricinus* ticks were positive for *R. monacensis* (11%) and *R. slovaca* (18.2%) [22]. However, all 89 tested blood samples yielded negative results [22]. Similarly, in Spain, *D. marginatus* ticks were positive for *R. slovaca* (7.3%) and *R. raoultii* (2.7%); *Rhipicephalus sanguineus* sensu lato for *R. massiliae* (13.7%); and *Hyalomma lusitanicum* for *R. slovaca* (0.3%), whereas all 167 spleen samples were negative [21].

Although these different results may be explained by a large variety of reasons, such as the geographical location of the study and tick populations and diversity, SFG rickettsiae prevalence in ticks and wild boar population characteristics, including the subspecies present, the detection protocol and the sample type, may also have an important influence. Based on the abovementioned studies, the most common positive types of samples seem to be ear and skin biopsies, followed by spleen and, less commonly, whole blood samples. Thus, blood samples may be considered less appropriate compared to skin biopsies. This could be due to short-lasting bacteremia in wild boars. Considering this, an advantage of biopsies could be the higher detection of pathogens. However, a limitation of this approach could be an overestimation of the pathogen pressure/transmission in the vector’s populations. On the other hand, spleen and blood samples indicate bacteremia, during which the pathogen transmission to the vectors is more important, but an underestimation of the prevalence may occur. Thus, the sample type could be chosen taking into account the study’s aim and proper discussions. Even though a moderate prevalence (4.4%) of SFG rickettsiae was observed in the present study, considering the possible short-lasting bacteremia, the sampling period and the absence of ticks on the animals, it would be expected to obtain even greater prevalence in the period of more intense activity of the tick vectors. 

*Rickettsia slovaca,* the most detected species in wild boar tissue samples, is mainly transmitted by *Dermacentor* species, while *Rickettsia monacensis* and *R. helvetica* (the two species detected in the present study) are transmitted primarily by *Ixodes ricinus* [39], a more generalist tick species, which is also the most common tick in Romania [40]. The activity peaks of *I. ricinus* are in May –June (as reviewed by [10]), with more reduced activity during the rest of the year. While this is the case with the abundant subadult stages, adults (although not recorded here) are active in the sampling period of the present study (October–February), too.

In Romania, nine *Rickettsia* species have been detected in ticks and animals so far (*R*. *aeschlimannii*, *R*. *conorii*, *R. felis*, *R. helvetica*, *R. hoogstraalii*, *R. massiliae*, *R. monacensis*, *R. raoultii* and *R. slovaca*) [41,42,43,44,45,46]. Among these, both *R. monacensis* and *R. helvetica* have a wide distribution, and they were detected all over Europe in ticks and other arthropods collected from different host species, as well as in hosts’ tissue samples [39]. 

*Rickettsia helvetica* has been reported in different tick species belonging to *Dermacentor*, *Ixodes*, *Haemaphysalis*, *Hyalomma* and *Rhipicephalus* genera, in unfed ticks or ticks collected from a large variety of hosts species [39]. In addition, *R. helvetica* has also been isolated in tissue samples from different animals, such as birds, bats, goats, mouflons, deer, mustelids, hedgehogs and rodents [39]. Similarly, *R. monacensis* has been isolated from tick species belonging to *Dermacentor*, *Ixodes*, *Haemaphysalis* and *Rhipicephalus* genera, from unfed ticks or ticks collected from a large variety of hosts species, and also from animal tissue samples, such as birds, bats, dogs and rodents [39,45,46].

In Romania, these rickettsial species were jointly detected, in the same geographical area, in *I. ricinus* as well as in *H. punctata,* both in free stages [42,43]; in ticks collected from different animal species [44,45]; and in *I. ricinus* populations collected from a wide array of hosts, including humans, dogs, cats, livestock, wild birds and mammals (for a comprehensive review, see [41]). Both *R. helvetica* and *R. monacensis* are recognized as zoonotic species, being reported as causing important illnesses, as described in several studies [47,48,49,50,51]. *R. monacensis* is the agent of MSF-like syndrome, which is characterized by non-specific flu-like symptoms, hypersensitivity skin reactions and rash. Although initially considered non-pathogenic, *R. helvetica* may cause fever and eczema. The resulting complications are in the form of subacute meningitis or myopericarditis. This infection has been implicated in cases of fetal pericarditis, sarcoidosis and febrile syndromes [47,48,49,50,51]. Their presence in wild boar blood observed in this study suggests that this species may be involved in different SFG *Rickettsia* spp. epidemiology in Europe.

The detection of *A. phagocytophilum* in wild boars was not unexpected. This bacterium was previously detected in Romanian boars [52]. Although the protocol used was the same (targeting the 16S rRNA), the prevalence obtained in this study (3%) was lower compared to the 4.48% prevalence in Transylvania previously published. However, in a previous report, the highest prevalence was obtained in Sibiu county, whereas wild boars sampled in the same (Cluj and Sălaj) counties were negative [52]. The previous study was conducted between 2007 and 2011, while the present study was conducted between 2019 and 2022, suggesting a possible increase in time for prevalence. Previous reports have suggested wild boar as a potential reservoir host for *A. phagocytophilum* [53]. The agent was detected in tissue samples from wild pigs in Belgium, the Czech Republic, Germany, Italy, Poland, Romania, Slovakia and Slovenia (as reviewed by [12]), as well as, recently, in Hungary [7]. It is the etiological agent of tick-borne fever and equine, canine and human granulocytic anaplasmosis. *Anaplasma phagocytophilum* is widely distributed across the whole of Europe, being identified in different tick and vertebrate hosts in at least 35 countries (as reviewed by [12]). The strains isolated from wild boars (based on more than one gene’s analysis) belong to the same cluster, ecotype and even genotype as the strains isolated from humans [36,43,54,55]. Thes BLAST analysis of the strains detected in the present study confirms their belonging to ecotype I [36].

## 5. Conclusions

In conclusion, the present study highlights the presence of three emerging zoonotic pathogens also in Romanian wild boars, reporting, for the first time, the presence of *R. monacensis* in the blood samples of these animals. The observed prevalence suggests the involvement of wild boars in the epidemiology of these rickettsiae species and *A. phagocytophilum*, and based on this, boars may be used as sentinels for the general surveillance of these zoonotic pathogens.

## Figures and Tables

**Figure 1 animals-13-01743-f001:**
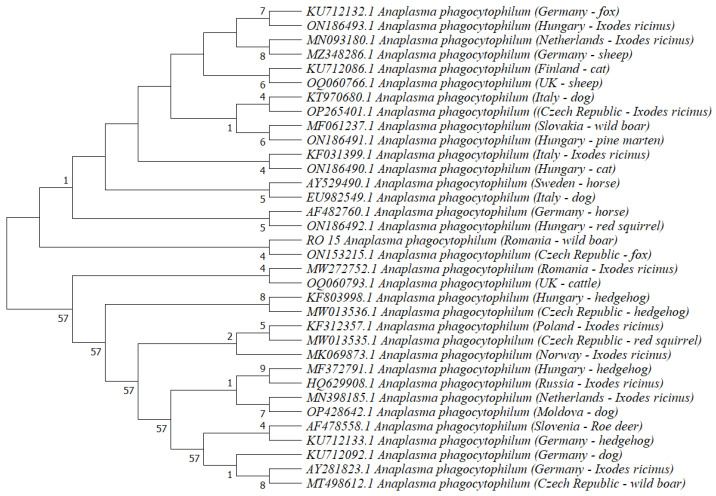
Phylogenetic tree based on *groEL* gene of *A. phagocytophilum*.

**Figure 2 animals-13-01743-f002:**
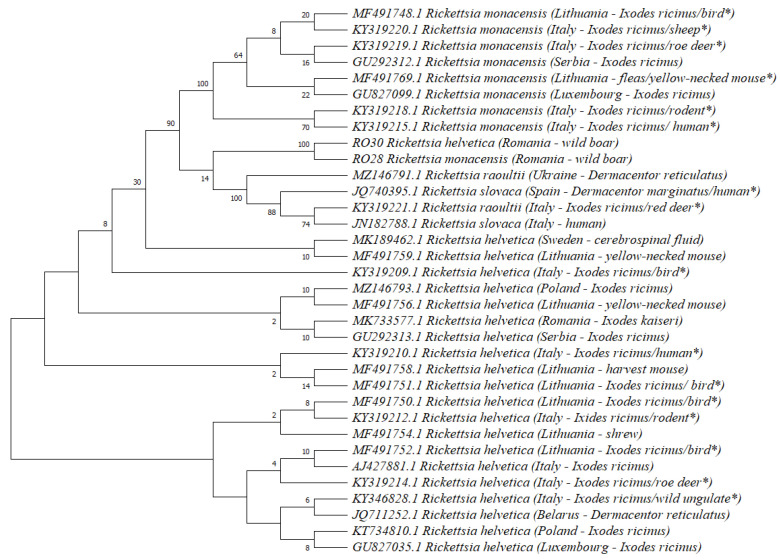
Phylogenetic tree based on *17 kDa* surface antigen coding gene of *Rickettsia* species distributed in Europe. *—isolates were obtained from the mentioned tick species collected from the mentioned hosts.

**Table 1 animals-13-01743-t001:** Primer sequences used for each pathogen and gene.

Pathogen	Primer Name and Sequence	Gene	Reference
*Anaplasma* spp./ *A. phagocytophilum*	Ge3a: CACATGCAAGTCGAACGGATTATTC Ge10: TTCCGTTAAGAAGGATCTAATCTCC Ge2: AACGGATTATTCTTTATAGCTTGCT Ge9: GGCAGTATTAAAAGCAGCTCCAGG	16S rRNA	[30]
*A. phagocytophilum*	EphplgroEL(569)F: ATGGTATGCAGTTTGATCGC EphplgroEL(1193)R: TCTACTCTGTCTTTGCGTTC EphgroEL(1142)R: TTGAGTACAGCAACACCACCGGAA	*groEL*	[31]
*Babesia/Theileria* spp.,	BJ1: GTCTTGTAATTGGAATGATGG BN2: TAGTTTATGGTTAGGACTACG	18S rRNA	[32]
*Borrelia* spp.	5SCB: GAGTTCGCGGGAGAGTAGGTTATTGCC 23SN: TCAGGGTACTTAGATGGTTCACTTCC	IGS	[33]
*Ehrlichia* spp./ *E. canis*	ECC: AGAACGAACGCTGGCGGCAAGCC ECB: CGTATTACCGCGGCTGCTGGCA Canis: CAATTATTTATAGCCTCTGGCTATAGGA HE3: TATAGGTACCGTCATTATCTTCCCTAT	16S rRNA	[34]
*SFG Rickettsia* spp.	17DrikP3 GGAACACTTCTTGGCGGTG 17DrikP2 CATTGTCCGTCAGGTTGGCG 17DrikP4 GAGAGATGCTTATGGTAAGAC 17DrikP5 GAGAGATGCTTATGGTAAGAC	17 kDA	[35]

## Data Availability

The data presented in this study are available in this article.

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
