# Peer review of "The Role of Wild Boars in the Circulation of Tick-Borne Pathogens: The First Evidence of Rickettsia monacensis Presence"

_animals, 2023, doi:10.3390/ani13111743_

Round 1

Reviewer 1 Report

Line 63. New Zealand

Line 86. Europe instead of Europa

Line 96. Anaplasma phagocytophilum instead of A. phagocytophilum.

Line 99. Babesia divergens instead of B. divergens.

Line 104. Spotted Fever Group Rickettsia instead of SFG Rickettsia.

Line 193. Rickettsia does not belong to Anaplasmataceae.

Line 195. phagocytophilum instead of phagocitophylum. 

The manuscript will benefit with the inclusion of a phylogenetic tree in the Results section comparing both R. monacensis and R. helvetica from other  studies.

I do not have any concern about the research conducted in this manuscript, but it needs extensive revision for language and grammar. I am also a non-English speaker but I am afraid it cannot be accepted in its current form. The authors excessively repeat words such as wild boars or altogether (spelled in different ways). I am aware that wild boars are the main focus of the research but it is not necessary to use up to nine times in 12 lines. 

Author Response

Dear Reviewer,

Thank you for your detailed review and suggestions. We have revised the manuscript by including phylogenetical trees for the both Rickettsia species, also for A. phagocytophilum as one of the Reviewers suggested. Also, the manuscript has been revised for English language and grammar.

REVIEWER 1

Reviewer 1 (R1): Comments and Suggestions for Authors

Line 63. New Zealand

Author response (AR): Corrected

R1: Line 86. Europe instead of Europa

AR: Corrected

R1: Line 96. Anaplasma phagocytophilum instead of A. phagocytophilum.

AR: Corrected

R1: Line 99. Babesia divergens instead of B. divergens.

AR: Corrected

R1: Line 104. Spotted Fever Group Rickettsia instead of SFG Rickettsia.

AR: Corrected

R1: Line 193. Rickettsia does not belong to Anaplasmataceae.

AR: yes, it is correct, the sentence was rewrite 

R1: Line 195. phagocytophilum instead of phagocitophylum.

AR: Corrected.

R1: The manuscript will benefit with the inclusion of a phylogenetic tree in the Results section comparing both R. monacensis and R. helvetica from other studies.

AR: We have performed phylogenetical analysis for the both Rickettsia species, also for A. phagocytophilum as suggested, and we have included in the manuscript.

R1: Comments on the Quality of English Language

I do not have any concern about the research conducted in this manuscript, but it needs extensive revision for language and grammar. I am also a non-English speaker but I am afraid it cannot be accepted in its current form. The authors excessively repeat words such as wild boars or altogether (spelled in different ways). I am aware that wild boars are the main focus of the research but it is not necessary to use up to nine times in 12 lines.

AR: The manuscript has been revised for English language and grammar.

Reviewer 2 Report

Dear authors,

The manuscript title is “The role of wild boars in the circulation of tick-borne pathogens: the first evidence of Rickettsia monacensis presence in wild boars and it aims to evaluate the presence of tick-borne pathogens in wild boar in Romania.

The topic falls within the aims and scope of the journal and it is important in the context of climate changes and urbanization of the wildlife. The abstract is clear and complete, the keywords are not repeated from the title as sometimes is seen, and the background fits the theme well.

In Materials and Methods the authors should explain better how did they do the sampling. Was it a convenience one? It looks like. Also, the authors must say how and where did they collect the blood sample.

In the Discussion, it should be noted at the outset that the authors are comparing results based on different type of samples and their limitations/advantages/disadvantages.

Some particular suggestions/comments will be done here:

- Line 25 – doble “be”, delete one please

- Line 138 – the supplementary file 1 is the manuscript itself, you must upload it again

- Line 161 – something is wrong with the numbers in the end of this line

- Lines 175/176 – Rickettsia was detected in 9 samples. Two in 2019/2020 + 4 in 2021/2022 = 6. Something is missing here.

- Lines 213/214 – sorry but I do not understand this statement as Spain and Italy do not even border!

- Line 225 – “and” is not in italic please

- Line 226 – here and before you have some citations which are not in the reference style of this journal, please check.

- Line 256 – please add “)”

- Line 183 – to my knowledge blood is not considered a tissue sample

Author Response

Dear Reviewer,

 Thank you for your detailed review and suggestions. We have revised the manuscript by including phylogenetical trees for the both Rickettsia species, also for A. phagocytophilum as one of the Reviewers suggested. Also, the manuscript has been revised for English language and grammar.

REVIEWER 2

Reviewer 2 (R2): Comments and Suggestions for Authors

Dear authors,

The manuscript title is “The role of wild boars in the circulation of tick-borne pathogens: the first evidence of Rickettsia monacensis presence in wild boars” and it aims to evaluate the presence of tick-borne pathogens in wild boar in Romania.

The topic falls within the aims and scope of the journal and it is important in the context of climate changes and urbanization of the wildlife. The abstract is clear and complete, the keywords are not repeated from the title as sometimes is seen, and the background fits the theme well.

In Materials and Methods, the authors should explain better how did they do the sampling. Was it a convenience one? It looks like. Also, the authors must say how and where did they collect the blood sample.

Author response (AR): Thank you for the positive feedback. Additional information on samples collection was added: in Salaj a single hunting area was considered, and the collected samples represent all the animals shouted in the study time period; moreover, in the third year we included samples from another county (Cluj) since in the first chosen location the wild boar’s population decreased considerably being affected by African swine fever virus.

R2: In the Discussion, it should be noted at the outset that the authors are comparing results based on different type of samples and their limitations/advantages/disadvantages.

            AR: Thank you for your remark, we tried to find possible explanation for the different results from different studies.

R2: Some particular suggestions/comments will be done here:

- Line 25 – doble “be”, delete one please

AR: Done.

R2: - Line 138 – the supplementary file 1 is the manuscript itself, you must upload it again

AR: We have included the table from the supplementary material in the manuscript, as table nr. 1.

R2: - Line 161 – something is wrong with the numbers in the end of this line

AR: We have revised the sentence.

R2: - Lines 175/176 – Rickettsia was detected in 9 samples. Two in 2019/2020 + 4 in 2021/2022 = 6. Something is missing here.

AR: Thank you, we have corrected (we missed the samples in Cluj (n=3).

R2: - Lines 213/214 – sorry but I do not understand this statement as Spain and Italy do not even border!

AR: It is true, but the two countries have in some part’s similar climate (Mediterranean influence) and similar tick’s and wild boar’s populations.

R2: - Line 225 – “and” is not in italic please

AR: Corrected

R2: - Line 226 – here and before you have some citations which are not in the reference style of this journal, please check.

AR: We have revised all the references.

R2: - Line 256 – please add “)”

AR: Corrected.

R2: - Line 283 – to my knowledge blood is not considered a tissue sample

AR: We have revised the sentence.

Reviewer 3 Report

After reading the manuscript entitled "The role of wild boars in the circulation of tick-borne pathogens: the first evidence of Rickettsia monacensis presence in wild boar" I would like to make two minor remarks:

1) It is not quite clear to me why the authors moved the sequence of the used oligonucleotides and the PCR conditions to a separate Supplementary file. This file is not that big and all the information could be included in the main text

2) I am somewhat confused by the phrase

"In the absence of the dominant sow (old matriarch), the group can travel up to 1000 ha in a day, sometimes reaching anthropic zones, including peri-urban areas in searching for food"

Hectares is a measure of area, and those are distances. In the link provided by the authors, I also could not find an explanation of this point.

Author Response

Dear Reviewer,

Thank you for your detailed review and suggestions. We have revised the manuscript by including phylogenetical trees for the both Rickettsia species, also for A. phagocytophilum as one of the Reviewers suggested. Also, the manuscript has been revised for English language and grammar.

REVIEWER 3

Reviewer 3 (R3): Comments and Suggestions for Authors

After reading the manuscript entitled "The role of wild boars in the circulation of tick-borne pathogens: the

first evidence of Rickettsia monacensis presence in wild boar" I would like to make two minor remarks:

R3: 1) It is not quite clear to me why the authors moved the sequence of the used oligonucleotides and the PCR conditions to a separate Supplementary file. This file is not that big and all the information could be included in the main text

            Author response (AR): We have included the table from the supplementary material in the manuscript, as table nr. 1.

R3: 2) I am somewhat confused by the phrase

"In the absence of the dominant sow (old matriarch), the group can travel up to 1000 ha in a day, sometimes reaching anthropic zones, including peri-urban areas in searching for food"

Hectares is a measure of area, and those are distances. In the link provided by the authors, I also could not find an explanation of this point.

            AR: Indeed, in the cited reference the author revised the home range size and they highlighted that the home ranges vary between “0.62 to 34.04 km2 for females and 1.05 to 48.34 km2 for males”, being an important difference between females and males. Thus, in the absence of the dominant sow the home range may increase. The additional information on the importance of the dominant sow in the group behavior was found in a hunting exam book but we didn’t used the citation since is in Romanian only (Åželaru N. Manual pentru examenul de vânător. Editura Cynegis, BucureÈ™ti. 2001.). We replaced the reference since in this book the biology, including the behavior of the species, is better described.

Round 2

Reviewer 2 Report

Dear authors,

Congratulations for the improvements!

However, I still miss this in discussion as asked before:

In the Discussion, it should be noted at the outset that the authors are comparing results based on different type of samples and their limitations/advantages/disadvantages.

Best regards

Author Response

Reviewer 2

Dear authors,

Congratulations for the improvements!

However, I still miss this in discussion as asked before:

In the Discussion, it should be noted at the outset that the authors are comparing results based on different type of samples and their limitations/advantages/disadvantages.

Author response: Thank you for your suggestion. We have added the paragraph: 'Although, these different results may be explained by a large variety of reasons such as: geographical location of the study, tick’s populations and diversity, SFG rickettsiae prevalence in ticks and the wild boars’ populations characteristics including the subspecies present; the detection protocol and the sample type may also have an important influence. Based on the above-mentioned studies, the most common positive type of samples seem to be ear and skin biopsy, followed by spleen and less commonly whole blood samples. Thus, the blood samples may be considered less appropriate compared to skin biopsy. This could be due to a short-lasting bacteremia in wild boars. Considering this, the advantages of the biopsies could be a higher detection of the pathogen. However, as a limitation of this approach could be an overestimation of the pathogen pressure/transmission in the vector’s populations. On the other hand, spleen and blood samples indicate a bacteremia during of which the pathogen transmission to the vectors is more important, but an underestimation of the prevalence may occur. Thus the sample type could be chosen taking into account the study’s aim and properly discussed.'